# Identity management in the face of HIV and intersecting stigmas: A metasynthesis of qualitative reports from sub-Saharan Africa

**Alanna J. Bergman**[1], **Katherine C. McNabb**[1]*, **Khaya Mlandu**[2], **Alvine Akumbom**[3], **Dalmacio Dennis Flores**[4]

**1** Center for Infectious Disease and Nursing Innovation, School of Nursing, Johns Hopkins University, Baltimore, Maryland, United States of America, **2** Izikhuba Unjani Clinic, Mngungundlovu, South Africa, **3** Johns Hopkins University School of Nursing, Baltimore, Maryland, United States of America, **4** University of Pennsylvania School of Nursing, Philadelphia, Pennsylvania, United States of America

* kmcnabb2@jhmi.edu

**Data Availability Statement:** All data underlying the findings are available in the paper or in the Supporting Information files.

## Abstract

While stigma experienced by people living with HIV (PLWH) is well documented, intersectional stigma and additional stigmatized identities have not received similar attention. The purpose of this metasynthesis is to identify salient stigmatized intersections and their impact on health outcomes in PLWH in sub-Saharan Africa. Using Sandelowski and Barroso's metasynthesis method, we searched four databases for peer-reviewed qualitative literature. Included studies (1) explored personal experiences with intersecting stigmas, (2) included ≥1 element of infectious disease stigma, and (3) were conducted in sub-Saharan Africa. Our multinational team extracted, aggregated, interpreted, and synthesized the findings. From 454 screened abstracts, the 34 studies included in this metasynthesis reported perspectives of at least 1258 participants (282 men, 557 women, and 109 unspecified gender) and key informants. From these studies, gender and HIV was the most salient stigmatized intersection, with HIV testing avoidance and HIV-status denial seemingly more common among men to preserve traditional masculine identity. HIV did not threaten female identity in the same way with women more willing to test for HIV, but at the risk of abandonment and withdrawal of financial support. To guard against status loss, men and women used performative behaviors to highlight positive qualities or minimize perceived negative attributes. These identity management practices ultimately shaped health behaviors and outcomes. From this metasynthesis, the Stigma Identity Framework was devised for framing identity and stigma management, focusing on role expectation and fulfillment. This framework illustrates how PLWH create, minimize, or emphasize other identity traits to safeguard against status loss and discrimination. Providers must acknowledge how stigmatization disrupts PLWH's ability to fit into social schemas and tailor care to individuals' unique intersecting identities. Economic security and safety should be considered in women's HIV care, while highlighting antiretrovirals' role in preserving strength and virility may improve care engagement among men.

**Funding:** AJB is supported by the Johns Hopkins University School of Nursing Discovery and Innovation Fund, and the Sigma/Association of Nurses in AIDS Care Research Grant. KM is supported by the National Institute of Allergy and Infectious Diseases (F30AI165167). The funders had no role in study design, data collection and analysis, decision to publish, or preparation of the manuscript.

**Competing interests:** The authors have declared that no competing interests exist.

## Introduction

Stigma associated with HIV is well described in the literature as a barrier to patient engagement throughout the care continuum. Stigma can be defined as a negatively perceived attribute that precludes an individual from full social acceptance and contributes to social, financial and/or health related inequity [1, 2]. Researchers have identified associations between HIV stigma and healthcare avoidance [3], poorer ART adherence [3–6], anxiety [7] and depression [4, 6–8]. However, HIV stigma is not the only type of stigma that individuals encounter in their daily lives [9, 10]. Each community has its own norms and values. Violations of these norms can, and often do, lead to stigma. In addition to the stigma of living with HIV, individuals are subject to a variety of other stigmas depending on the cultural context, including but not limited to gender stigma, sexual identity stigma, mental health and substance use stigma, and stigma related to other communicable illnesses. People may encounter stigma in different degrees or in variable qualities due to particular combinations of stigmatized identities. Intersectional stigma is the convergence of more than one of these stigmatized identities. Intersectional stigma acknowledges that living with multiple stigmatized identities does not simply create an additive effect; the negative ramifications of intersectional stigma can be multiplicative or exponential [11]. Intersectional stigma also accounts for factors that ease or mitigate stigma [11].

Infectious disease stigma stemming in part from fear of contagion [12, 13] is well documented across history, and is codified in biblical references; but the HIV epidemic introduced a new facet of infectious disease stigma. Because of its association with sexual behavior and injection drug use, HIV stigma became entwined with questions of morality, invoking stereotypes and debates about blame [14]. Financial loss, religious beliefs, and other institutional authoritarianism can also precede community level stigma impacting internalized, perceived, and experienced stigma at the individual level [1]. Internalized stigma refers to how individuals see themselves as a result of stigma. Perceived stigma references how an individual believes the wider community views members of the stigmatized group. Finally, negative attitudes, insults, and discrimination experienced by the stigmatized person is commonly called enacted stigma [1, 15].

Stigmatization is a societal mechanism for the differentiation and negative stereotyping of people living with stigmatized identities which then leads to their poorer quality of life [16]. Explanations of social categorization and intergroup behavior contribute additional context for our interpretations of stigma. Humans take pride in, and extract meaning from social inclusion; each person situates themselves and others into groups that help to explain belonging and identity [1, 2, 16, 17]. These context dependent social categories often lead to conflict between groups in an effort to ascribe positive attributes to one's own social schema resulting in negative reflection of other social groups [15–17]. As a result, salient intersections contributing to stigma vary widely across geography, culture, and community.

Reducing infectious disease stigma is a public health priority acknowledged by both the World Health Organization and the Joint United Nations Programme on HIV/AIDS [18, 19]. Because intersectional stigma may include a complex combination of identities and attributes that contribute both intensifying and mitigating factors, intersectional stigma is difficult to quantify. To date, there is no validated measure of intersectional stigma. Qualitative research methodologies allow rigorous exploration of intersectional stigma consistent with the theoretical underpinnings of the construct. Qualitative research will help to inform the scope and impact of intersectional infectious disease stigma, and future interventions towards reducing these stigmas.

The purpose of qualitative meta-synthesis is to bridge the gap between individual studies and to identify broad truths between the lived experiences of individuals [20]. Although intersectional infectious disease stigma in the United States and Canada is well studied [5, 21–23],

it is less studied in sub-Saharan Africa. Twenty-two of the top 30 high burden TB/HIV countries are located in Sub-Saharan Africa [24]. Therefore, our primary purpose in conducting this review was to synthesize the current body of qualitative evidence in order to determine the intersections that impact infectious disease stigma in sub-Saharan Africa and consider future stigma mitigation strategies.

## Methods

Qualitative meta-synthesis is a methodic and rigorous process to identify, abstract and synthesize qualitative data pertinent to a specific research question [20]. We used the process outlined by Sandelowski and Barroso to guide our methods [20]. A thorough search of the literature was conducted in conjunction with a library informationist across four databases, Medline, CINAHL, PsychINFO, and African Index Medicus. Because intersectional stigma is a relatively new phenomenon of interest with broad definitions, we used a combination of search strategies to maximize recall rather than precision seeking the greatest number of relevant documents [20]. Search terms for intersectional stigma included: intersectional stigma; double stigma; layered stigma, and multiple stigmas. Following PRISMA guidelines, two independent reviewers initially performed title and abstract screening on all studies, and then moved to full text screening, discarding studies that were irrelevant to our study purpose. Inclusion criteria were: studies conducted in sub-Saharan Africa, performed an analysis of intersecting stigmas, explored the experiences of people impacted by stigma [through experiential or key informant testimony], included at least one element of infectious disease stigma [HIV, TB, hepatitis B, hepatitis C, human papilloma virus etc.] and had a qualitative approach in the study design. The infectious diseases included in the search strategy were chosen based on geographic disease burden and the stigma literature. We aimed for an adaptable interpretation of intersectional stigma to include studies that contained thematically relevant findings even if the main study purposes were not focused on stigma as a main outcome [20]. We did not exclude studies by publication date or qualitative methodology.

Each included study was imported into F4analyse (Marburg, Germany), a computer-assisted qualitative data analysis software which was used to store and organize data. Together the two reviewers double coded five randomly selected studies to develop an initial codebook. Each reviewer was then assigned additional articles to code independently. Every fifth study was double coded to ensure congruence. Reviewers independently memo-ed reflecting on emerging themes and possible biases. The coders met weekly with a South African researcher to discuss the analytic process and consider varying viewpoints with a cultural insider. New codes were added iteratively. The team used elements of reciprocal translation, imported concepts and use of event timelines to group codes and develop larger themes from the individual studies [20]. After extracting the main themes, the study team developed a conceptual model that captured the key elements of the synthesis.

The team consisted of researchers from diverse racial and national/ethnic backgrounds. The strength of this diversity is that several of the marginalized groups included in the studies were represented on the research team. This allowed us to constantly reflect on our background, positionality, and privilege given our concomitant roles as emic and etic researchers throughout the review. Team members from South Africa and Cameroon allowed us to consider cultural viewpoints, clarify language and interrogate western bias in global research. The methods expert guiding the analysis and synthesis of the project is a non-White male researcher born outside of the United States with expertise in qualitative meta-synthesis. Collectively, our team has a longstanding track-record of working with PLWH, marginalized groups and in global health work.

## Results

The search generated an initial 454 studies for title and abstract review. Forty-nine studies moved on to full text screening. References from included studies were reviewed to identify potentially relevant research that was not found in the initial search. We also reviewed recent publications from the work of global stigma researchers to identify studies not yet indexed. Through this process, we identified two additional papers that met our inclusion criteria. These papers were imported directly into full text review and subsequently included. Ultimately, 34 studies were included for full abstraction and analysis (See Fig 1).

Most of the included studies were conducted in either South Africa (n = 10) or Uganda (n = 9). Three studies were conducted in Malawi, two in Botswana and individual studies took place in Cameroon, Ethiopia, Ghana, Kenya, Nigeria, Rwanda, Swaziland, Tanzania, Zambia, and Zimbabwe. This meta-synthesis represents the experiences and perspectives of at least of 1,228 individuals from sub-Saharan Africa. Of these, 918 were people testifying to the lived experience of stigma. An additional 310 were key informants that included healthcare workers, program administrators and care givers who reported on internalized, anticipated, and experienced stigma from an observers' perspective (See Table 1).

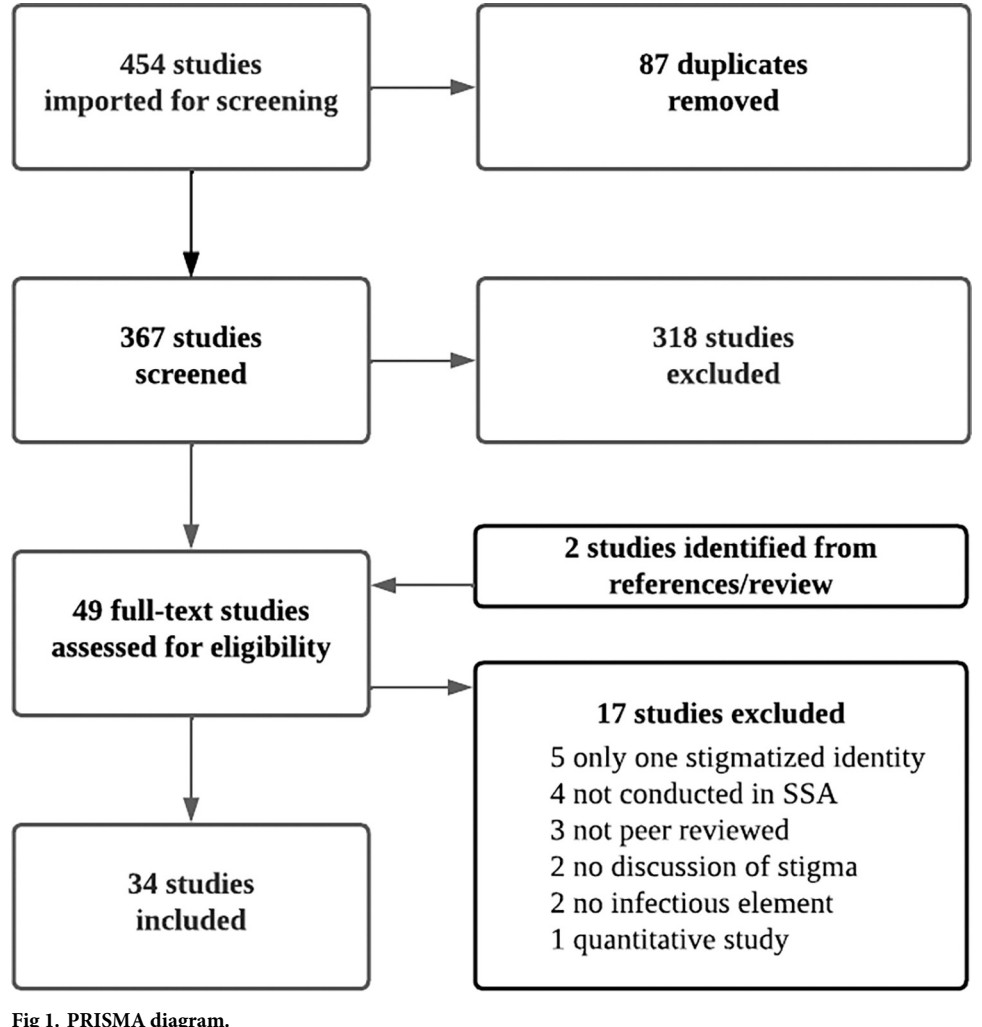

**Fig 1. PRISMA diagram.**

**Table 1. Participant demographics.**

| Type of participant | N (%) |
|---|---|
| Experiential participants | 948 (75%) |
| • Female | 557 (59%) |
| • Male | 282 (30%) |
| • Unknown gender | 109 (11%) |
| Key informants | 310 (25%) |
| Total | **1258** |

All of the studies included HIV as a stigmatized identity in their intersectional analysis. The intersections of interest included: tuberculosis and HIV (8), gender and HIV (6), mental health and HIV (5), older age and HIV (3), non-communicable disease such as hypertension or diabetes and HIV (3), lesbian, gay, bisexual, trans, and queer (LGBTQ) communities and HIV (3), substance use and HIV (2), occupational nursing and HIV (2), refugee status and HIV (1) and one study looked at race, socioeconomic status, and HIV. These intersections are shown in S1 Table. Studies that investigated gender as a stigmatized identity did so in the context of gender-based violence, pregnancy, and by comparing the experiences of cis-gender women to that of cis-gender men (intra-categorical stigma) [25]. Studies that evaluated the experiences of HIV within the LBGTQ community included men who have sex with men (MSM) and transgender women. Taken together, the included studies offer an explanatory model for framing identity and stigma in sub-Saharan Africa focused on role expectation and fulfillment. Table 2 shows the characteristics of the included studies.

## Conceptual framework

The Stigma Identity Framework captures (Fig 2) the major themes and their inter-relationship. The framework fits together elements of intersectionality, stigma theory and identity theory in order to holistically depict the experiences of the people in these studies. It also illustrates how affected individuals may minimize or emphasize certain traits and identities to protect themselves against status loss and discrimination. Further, the framework lays out the intersectional identities from the included studies and how these identities impact an individual's capacity to fulfill their expected social roles. Depending on the community and prevailing value systems, individual identity management may focus on the roles of a spouse or parent, level of productivity, extent of community engagement, and other culturally defined identities. The ability to fully engage in these social roles is impacted by disclosure, or community awareness of "undesirable" traits or identities. In order to protect themselves from being outed or discriminated against, many individuals endorsed performative identities to highlight positive qualities or to minimize perceived negative attributes [26]. The extent to which an individual can balance their expected social roles by preventing disclosure, and through identity management, impacts healthcare related behaviors and consequent health outcomes. The thematic results that follow are grouped into (1) *identity and visibility disclosure*, (2) *ability to fulfill expected social roles* and (3) *performative identities*. The ways that individuals cannot, or do not navigate identity, disclosure, and social fulfillment through performative action impact their experience of intersectional stigma and healthcare behaviors which dictate infectious disease outcome.

## Ability to fulfill expected social roles

**Adult identity and productivity.**   Across studies, the role of the adult was closely tied to social and financial productivity and familial/community responsibilities. All participants

**Table 2. Characteristics of included studies.**

| First author & year | Country | Stigma is study objective (Y/N) | Intersection of interest | Conceptual framework | Number and type of participant | Experiential or key informant |
|---|---|---|---|---|---|---|
| **Angwenyi 2018** [38] | Malawi | N–purpose was to examine patient self-management of HIV and other chronic illnesses | HIV + non-communicable disease (hypertension, epilepsy, stroke, asthma etc). | Bandura's theory of self-efficacy | 14 in-depth interviews 33 focus group participants | Experiential participants only |
| **Becker 2019** [39] | Botswana | Y | HIV + mental health | Kleinman's explanatory model Link & Phelan's theory of stigma | 42 in-depth interviews | Both |
| **Brown 2018** [40] | South Africa | N–study purpose was to develop a curriculum for intersectional HIV education | HIV + race + socioeconomic status | Intersectionality theory | 86 participants using photovoice, narratives, drawing and self-reflective assignments | Key informants only |
| **Buregyeya 2012** [41] | Uganda | N–explore healthcare workers utilization of occupational TB & HIV services | HIV + TB | None identified | 8 focus groups (total number of participants not reported) | Experiential participants only |
| **Chileshe 2010** [42] | Zambia | N–understand experiences of antiretroviral therapy access for people living with TB and HIV | HIV + TB | None identified | 9 index participants and their households' using anthropological assessments | Both |
| **Crankshaw 2014** [43] | South Africa | N–study objective was to explore disclosure | HIV + unintentional pregnancy | None identified | 62 in-depth interviews | Experiential participants only |
| **Daftary 2007** [44] | South Africa | N–explore TB status disclosure and the decision-making process for HIV testing acceptance or refusal | HIV + TB | None identified | 21 in-depth interviews | Experiential participants only |
| **Daftary 2012a** [45] | South Africa | Y | HIV + TB | Stigma theory from Goffman, Phelan & Link and Farmer | 40 in-depth interviews | Experiential participants only |
| **Daftary 2012b** [46] | South Africa | N–study looked at care seeking among people living with TB and HIV | HIV + TB | None identified | 40 in-depth interviews | Experiential participants only |
| **Ezeanolue 2020** [47] | Nigeria | N–explored barriers to integrating mental health care with HIV care | HIV + mental health | None identified | 80 focus group participants | Key informants only |
| **Finnie 2010** [48] | South Africa | N–study explored perceptions of TB and TB care-seeking | HIV + TB | None identified | 12 in-depth interviews | Key informants only |
| **Freeman 2017** [49] | Malawi | N–study explored identity among older adults living with HIV | HIV + aging | Burke's identity control theory and interactionist framework | 43 in depth interview 30–45 focus group participants | Experiential participants only |
| **Gebremariam 2010** [50] | Ethiopia | Y | HIV + TB | Cumings theoretical framework (1980). | 24 in-depth interviews 14 focus group participants | Both |
| **Gnauck 2013** [51] | Kenya | Y | HIV + gender | None identified | 60 focus group participants | Experiential participants only |
| **Jani 2021** [52] | Tanzania | Y | HIV prevention + gender | Framework for PrEP introduction for adolescent girls and young women | 28 in-depth interviews | Experiential participants only |
| **Kellett 2016** [53] | Uganda | Y | HIV + gender | Tsai, Bangsberg & Weiser's conceptualization of HIV stigma | 54 focus group participants | Experiential participants only |

*(Continued)*

**Table 2.** (Continued)

| First author & year | Country | Stigma is study objective (Y/N) | Intersection of interest | Conceptual framework | Number and type of participant | Experiential or key informant |
|---|---|---|---|---|---|---|
| **Kennedy 2013** [54] | Swaziland | N–understand the health, dignity and prevention needs of MSM living with HIV | HIV + LGBTQ (MSM) | Positive health, dignity and prevention framework | 62 participants in focus groups and in-depth interviews | Both |
| **King 2019** [55] | Uganda | N–explore gender identity and expression as they relate to HIV risk, healthcare seeking and STI prevention | HIV + LGBTQ (transgender women) | Syndemic theory on gender identity, HIV, stigma and social determinants of health | 45 in-depth interviews | Experiential participants only |
| **Kuteesa 2012** [56] | Uganda | Y | HIV + aging | None identified | 40 in-depth interviews and focus group participants | Experiential participants only |
| **Kyakuwa 2009** [57] | Uganda | Y | HIV + nurses | None identified | 6 Nurses living with HIV were interviewed using anthropological methods [life histories, observation, informal conversation, diary analysis etc]. | Experiential participants only |
| **Kyakuwa 2012** [58] | Uganda | N–explore the relationship between HIV expert clients and HIV nurses | HIV + nurses/ healthcare workers | None identified | 67 in-depth interviews and anthropological assessments | Both |
| **Lemasters 2020** [59] | Malawi | N–study explored experiences of post-natal depression among women living with HIV | HIV + mental health (post-natal depression) | None identified | 24 in-depth interviews | Experiential participants only |
| **Logie 2021** [60] | Uganda | N–principally studied HIV self-testing among urban refugee youth | HIV testing + refugee status | Health Stigma and Discrimination Framework | 5 focus group discussions | Both |
| **Magidson 2019** [61] | South Africa | N–this was a qualitative implementation study used to tailor an intervention for HIV and substance use disorder treatment integration | HIV + substance use | RE-AIM framework | 30 in-depth interviews | Both |
| **Matima 2018** [62] | South Africa | N–study looked at patient experiences with HIV/DMII multi-morbidity | HIV + Non communicable disease (diabetes mellitus II) | Cumulative complexity model | 16 in-depth interviews | Both |
| **Matlho 2017** [63] | Botswana | N–develop an understanding of barriers and facilitators towards HIV initiatives tailored for older adults | HIV + aging | Shiffman and Smith's framework on determinants of political priority for global initiatives | 15 in-depth interviews | Key informants only |
| **Mburu 2014** [64] | Uganda | N–explore community-based peer support groups and engagement in peer HIV support | HIV + gender | Wyrod's framework Intersectionality theory | 25 in-depth interviews 40 focus group participants | Both |
| **Mugisha 2020** [65] | Uganda | N–explored barriers to HIV care engagement from the provider perspective | HIV + mental health | None identified | 15 in-depth interviews | Key informants only |
| **Njozing 2010** [66] | Cameroon | N–understand barriers and facilitators of HIV testing among people living with TB | HIV + TB | None identified | 12 in-depth interviews | Experiential participants only |
| **Owusu 2020** [67] | Ghana | N–study analyzed experiences of HIV along gender lines | HIV + gender | None identified | 38 in-depth interviews | Experiential participants only |
| **Regenauer 2020** [69] | South Africa | Y | HIV + substance use | Intersectional stigma framework by Bowleg & Turan | 30 in-depth interviews | Both |

*(Continued)*

**Table 2.** (Continued)

| First author & year | Country | Stigma is study objective (Y/N) | Intersection of interest | Conceptual framework | Number and type of participant | Experiential or key informant |
|---|---|---|---|---|---|---|
| **Russel 2016** [68] | Rwanda | N–describe trauma [gender-based violence and HIV infection] experienced by women due to conflict | HIV + gender (Gender-based violence) | None identified | 22 in-depth interviews | Experiential participants only |
| **Tokwe 2020** [70] | South Africa | N–aimed to explore experiences of living with HIV and hypertension | HIV + non-communicable disease (HTN) | None identified | 9 in-depth interviews | Experiential participants only |
| **Tsang 2019** [71] | Zimbabwe | Y | HIV + LGBTQ (MSM) | Scambler's sociological perspective | 15 in-depth interviews | Experiential participants only |

highlighted a loss of productivity related to HIV and TB, as illness decreased their stamina and strength. This loss of productivity was perceived to undermine their value as productive adults. Many respondents were agricultural workers or walked long distances for work and found that fatigue, weight loss and generalized weakness decreased their strength or endurance. Across studies, participants repeatedly discussed loss of strength as a threat to their identity as productive adults.

Older adults articulated feelings of adult identity loss tied to lost productivity and the physical limitations of age. Despite the respect gained with age, there was a common perception that aging reduced an individual's ability to contribute to self and society. This was compounded by a diagnosis of chronic disease and appeared particularly acute for those suffering from symptomatic illness. Most participants reported pride in their work, their ability to maintain and provide for themselves, and wished to contribute to their communities. People who were unable to support themselves, required financial support from others, and who were unable to engage in self-care activities qualified their lapses in productivity as temporary setbacks surmountable with hard work and treatment. Framing loss of strength as a temporary condition allowed identity preservation as a return to productivity remained within reach.

## Gendered identity

Across all studies, societal role expectations differed by gender. Participants described males as financial providers. Men were expected to deliver economic support which validated their role as primary decision makers, including family healthcare decisions. Masculinity was tied to personal attributes and characteristics such as physical strength, rationality, and dominance. The chronic illness of HIV, TB, mental illness or noncommunicable disease, was incompatible with masculine cultural expectations as it rendered men weak, reliant on the care and support of others, and sidelined their familial authority. Similarly, the sexual behaviors of MSM challenged the ideals of African masculinity. MSM discussed enormous familial and community pressure to marry and have children which was inconsonant with their actual attractions, behaviors, or identities. Some cited the idea that homosexuality was unnatural to Africans and that same sex behaviors resulted from Eurocentric influence. In this way, same sex behaviors were perceived as a challenge not only to masculinity but to African identity.

In contrast to masculinity, which was tied to specific characteristics or personality traits, femininity and womanhood were tied to childbearing and household maintenance, rather than character or selfhood. During periods of upheaval or stress, women were viewed as emotional supports for all members of the family. Many male and female participants endorsed the idea of the physical subordination of women to men. Male respondents often viewed women

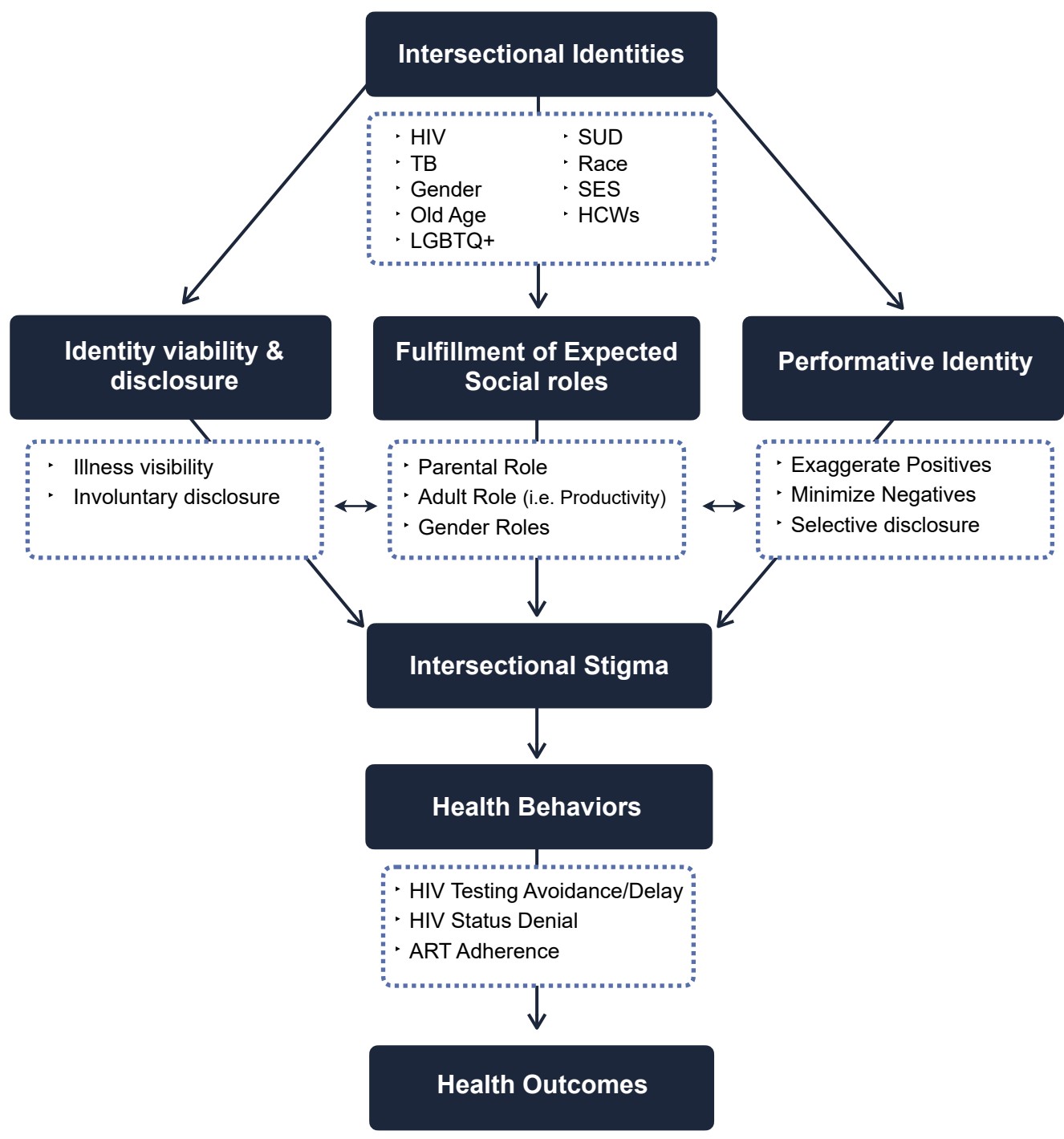

**Fig 2. Stigma identity framework.**

as the weaker sex and many women conceded that men were physically stronger and better equipped for manual labor than women. As a result of their social roles within the home, women were generally financially dependent on their male partners increasing risk of volatility and economic precarity.

### Parenting identity

Several of the included groups reported challenges to parenting. Individuals who felt unable to fulfill their expected roles as mothers or fathers deeply internalized and perceived social inadequacy. Both men and women considered childbearing and childrearing as key elements of their social identity. For men, parenthood affirmed their virility and solidified their role as household heads. For women, childbearing was key to their identity as women and as traditional caregivers. Because so much of the female identity was tied to maternal duties, women who experienced attacks to socially constructed ideas of motherhood felt acute identity violations. Women living with HIV who became pregnant worried about the health of their unborn infants and felt torn between their identity as mothers, and their identity as women living with chronic illness. Similarly, women who experienced post-natal depression deeply internalized their mental illness as a role violation unbecoming of a mother. Conversely, for men who endorsed same-sex behaviors, their sexuality was a barrier to traditional marriage and procreation. Many of the men interviewed were Christian and felt that their sexual behaviors and consequent inability to have children also violated religious expectations.

### Identity visibility and involuntary disclosure

Despite best efforts to maintain daily routines and a semblance of normalcy throughout infectious disease care and treatment, participants experienced symptomology or confidentiality betrayals that disclosed their identity status. Unlike people who chose to voluntarily disclose their identities to access support or camaraderie, others experienced involuntary disclosure or outing.

### Symptom visibility

Illness visibility was central to participant fears of disclosure and outing. Individuals who perceived that their physical presentation betrayed their disease status reported an additional burden of stigma and shame. People living with stigmatized illnesses felt that their bodies revealed their illness identities to the community. Many participants viewed weight loss as an indicator of infectious disease that would then cause the community to make assumptions or gossip about their health in general or HIV status in particular. Others reported that the audible and persistent TB cough, or visible lymphadenopathy alerted others to their disease status. PLWH also reported perceived and experienced ostracization–others stared, gossiped, insulted them, and questioned their HIV status making them feel despondent or hopeless. Particularly in South Africa, similar symptomology led to a conflation of HIV with TB, due to overlapping symptoms of the two disease processes. People living with TB reported that community members spread rumors that they were HIV-positive, regardless of their true HIV status.

### Visibility and outing via healthcare

The threat of being seen at HIV and TB clinics raised concerns among participants about privacy, confidentiality, and disclosure. People living with HIV and/or TB feared status disclosure that might invite speculation, physical violence, or abandonment. In most cases, individuals carefully safeguarded their HIV and TB status unless it was absolutely necessary to disclose. Therefore, being seen at infectious disease clinics was strictly avoided to prevent involuntary disclosure. In many cases, people were willing to travel great distances to seek treatment outside of their communities or in places where they felt certain that no one would recognize them. This presented significant barriers to treatment access and adherence.

Study participants who endorsed stigmatized identities or behaviors attempted to withhold pertinent information from healthcare providers. People who used drugs or alcohol frequently hid their substance use behaviors from healthcare providers. This omission was intended to maintain amicable therapeutic relationships with healthcare providers and to secure non-judgmental treatment for other healthcare needs. Similarly, MSM hid their sexual identities or sexual behaviors from healthcare providers believing that they would receive better care without full disclosure. in fact, some MSM felt more comfortable receiving care at foreign funded and staffed non-governmental organizations whom they believed were more accepting of same sex behaviors.

## Performative identities

Although stigma complicated role and identity fulfillment, many people living with stigmatized identities were able to maintain their social roles. Those who maintained their outward facing identities often escaped critique and judgement from others. In order to safeguard themselves from identity violations and stigma, individuals went to great lengths to carefully cultivate and project curated identities that endorsed assimilation within the larger group and thus normalcy.

## Selective disclosure

Throughout the articles, tuberculosis was inextricably intertwined with HIV. However, TB is not as heavily stigmatized as HIV in sub-Saharan Africa. Despite their different pathophysiology, and potential for cure, the TB and HIV share some of the same signs and symptoms; as a result, many PLWH used their TB status to explain physical changes to their health and appearance. When others pointed out noticeable weight changes, people living with both TB and HIV were more willing to disclose their TB status in order to satisfy curiosity. While TB was understood as easily transmissible and non-selective, HIV was perceived as preventable via "good" behavior. Therefore, TB disclosure had few implications for an individual's character and thus elicited limited or manageable blame. This was an explanatory narrative that could reasonably quell suspicion while maintaining a more desirable social identity.

Selective TB disclosure was more frequently reported by women. Women needed to disclose their TB status in order to secure financial support for treatment and travel but feared abandonment by their partners or families if they also disclosed their HIV status. For many female participants, a full figure indicated health and prosperity. Selective or partial disclosure of their TB diagnosis allowed women to address violations of societal beauty standards without the unrelenting stigma associated with HIV. According to participants, HIV and TB co-infection led to a tightrope walk between acceptable levels of scrutiny and too much disclosure in order to maintain safe housing and financial stability.

When participants were willing to share a stigmatized identity, they chose family or friends who were most likely to respect privacy and offer support. Often this included others who belonged to the same social identity group. MSM disclosed their identity to others within the LGBTQ community rather than family. Similarly, PLWH disclosed to others within treatment support groups. When disclosing to family or friends, individuals generally chose to disclose to female, rather than male, confidantes who were perceived as more empathetic which is also inline with traditional female identity roles.

## Sexual behaviors

Throughout these articles, we found evidence that men used sexual behavior to assert dominance and reinforce masculine ideals. As noted earlier, illness narratives arising from

patriarchal social norms challenged traditional notions of masculinity. In attempts to counter challenges to their masculinity, men wanted to cement their virility through sex by engaging with multiple sexual partners or being in extramarital relationships. Some men asserted sexual dominance by making unilateral decisions about condom use and sexual health, denying female input. By leaning into traditional masculine sexual narratives, men could minimize threats to their identity. Although women generally did not condone this hypersexual behavior, they did concede that it was an expected part of the male role which they tolerated. MSM also used traditional masculine narratives for identity protection. MSM participants endorsed the idea of heterosexual intercourse, marriage, and reproduction as a way of pacifying family and friends. By engaging in relationships with women, MSM could maintain a veneer of masculine heteronormativity.

In contrast to the exaggerated sexual behaviors of young men, older adults minimized their sexual activity in order to reduce the blame and culpability of living with HIV. There was a perception among key stakeholders that as people, and women in particular age, they lose their sexual drive, drastically reducing their risk for HIV infection. Although older adults revealed during confidential interviews that they had active, fulfilling sexual lives, they allowed a narrative of asexuality to continue in public spaces. Older adults were thought to be too well informed and self-disciplined to become infected with HIV through casual or unprotected sex. Older individuals did not challenge this narrative as it portrayed them in a more positive light.

## Healthcare behaviors

Among men, HIV and TB testing avoidance was common. Rather than subject their masculinity to existential threats, many men preferred not to know their disease status. Male participants living with HIV or TB recalled anxiety related to testing and reported that they delayed testing until their health was poor and they imminently feared death. This delay in testing led to longer recovery times, prolonged time away from work and greater physical dependence on others. Women also feared, and sometimes avoided, HIV testing. However, women's fears were tied to fears of violence and abandonment. Several included men also reported denial following diagnosis, which was not reported by women. In one article, HIV testing behaviors were hindered by a belief that HIV infection was primarily a Ugandan problem. As a result, key populations within the sampled refugee community believed that they were only at risk for HIV if they were associating with, and adopting the behaviors of, their Ugandan hosts. HIV testing was therefore dismissed as unnecessary.

Lastly, across studies, illness visibility consistently threatened status disclosure and caused much distress to participants. However, women reported the physical and aesthetic benefits of treatment adherence to their HIV and TB regimens. Adherence benefits such as weight gain, quelled cough, or reduction in lymphadenopathy was viewed as a central part of stigma prevention. Stigma mediated through illness identity and gender was a key driver of treatment adherence. Women also discussed the support that they received from other women through support groups, and empowerment projects. However, this finding was unique to women as men were reported to infrequently join or utilize support groups despite invitations and encouragement. Table 3 shows the thematic results by included studies.

## Discussion

This metasynthesis highlights the variety and scope of identities that contribute to intersectional stigma in Sub-Saharan Africa. Despite a clear focus on HIV stigma, other factors deemed culturally undesirable such as TB status, substance use disorder, and sexual identity contribute to stigma. Additional factors such as gender and age also interact with stigmatized

**Table 3. Thematic results by included study.**

| First author and year | Fulfillment of Expected Social Roles | | | Identity Visibility & Disclosure | | Performative Identity | | |
|---|---|---|---|---|---|---|---|---|
| | Adult Role | Gender Roles | Parental Role | Illness Visibility | Involuntary Disclosure | Selective Disclosure | Minimize Negative | Exaggerate Positive |
| Angwenyi 2018 [38] | X | | | | | | | |
| Becker 2019 [39] | | X | | | | | | |
| Brown 2018 [40] | | | | | | | | |
| Buregyeya 2012 [41] | | | | | X | | | |
| Chileshe 2010 [42] | | X | | X | X | X | | |
| Crankshaw 2014 [43] | | X | X | | | X | | |
| Daftary 2007 [44] | | | | X | | X | X | X |
| Daftary 2012a [45] | X | | | X | | X | X | X |
| Daftary 2012b [46] | | | | X | | X | | |
| Ezeanolue 2020 [47] | | | | | | | | |
| Finnie 2010 [48] | | | | X | | | | |
| Freeman 2017 [49] | X | | | X | | | X | X |
| Gebremariam 2010 [50] | | | | | | X | | |
| Gnauck 2013 [51] | | X | | | | | | |
| Jani 2021 [52] | X | | | | | | | |
| Kellett 2016 [53] | | X | X | | X | | | |
| Kennedy 2013 [54] | | X | | | X | X | X | X |
| King 2019 [55] | | X | | X | | X | | |
| Kuteesa 2012 [56] | X | | | X | | | | |
| Kyakuwa 2009 [57] | | | | | X | | | |
| Kyakuwa 2012 [58] | | | | | X | | | |
| Lemasters 2020 [59] | | | X | | | | | |
| Logie 2021 [60] | | X | | X | X | X | X | X |
| Magidson 2019 [61] | | | | | | | | |
| Matima 2018 [62] | | | | | | X | | |
| Matlho 2017 [63] | X | | | | | | X | X |
| Mburu 2014 [64] | | X | | | | | X | X |
| Mugisha 2020 [65] | | | | | | | | |
| Njozing 2010 [66] | | | | X | | | X | X |
| Owusu 2020 [67] | | X | | X | | X | | |
| Regenauer 2020 [69] | | | | | | | | |
| Russel 2016 [68] | | X | | | | | | |
| Tokwe 2020 [70] | | | | | X | | | |
| Tsang 2019 [71] | | X | X | | | | X | X |

identities to impact degrees of stigma and its manifestations. These themes are less salient in individual studies but become clear after reviewing the totality of intersectional work conducted in Sub-Saharan Africa.

Stigma is a gendered experience. This analysis underscores how both men and women from diverse settings within sub-Saharan Africa experience distinct, yet significant levels of stigma related to their infectious disease identities. Patriarchal institutions and beliefs around gender roles are harmful to women but are also deleterious to men. We do not believe that a universal African experience exists; and one is certainly not evident from the findings of this analysis. However, traditional masculine standards permeate across the included studies and result in the reported narratives of stigma. Because men in sub-Saharan Africa traditionally

play a key role in government, policy, community leadership and family structure, hegemonic masculinity also heavily dictates social constructs and definitions of normalcy [27, 28]. Rigid views of manhood in the context of sub-Saharan Africa create a constructed masculinity that, when threatened by a stigmatized illness result in negative coping behaviors and poor health outcomes. This fragile masculinity causes an identity crisis among men when they feel that due to perceived, experienced, or internalized stigma, they are unable to fulfill their role as household and community leaders. Women on the other hand more frequently experience stigma in the form of isolation and abandonment resulting in poverty. Women's economic reliance on men and the resulting financial precarity feeds back into a syndemic loop of illness and gendered vulnerability. As women experience abandonment and withdrawal of financial support, they experience barriers to health and well-being. Women also experience gender-based violence leading to emotional and physical vulnerability and lack the social and economic supports to improve their circumstances [29–31]. These factors put women at increased risk for poorer disease management thus reinforcing misconceptions about gendered fragility and subordination.

Despite a traditional focus on women from within intersectional analysis, our work shows that men also perceive and internalize stigma related to a variety of identities. Due to their ascribed roles as providers and community leaders, anticipated stigmatization presents a status threat for men that consequently undermines care engagement behaviors. While not intended to undermine or diminish the experiences of women who face disproportionate financial instability, inequity, hostility, and violence based on their gender, contrasting the experiences of men and women helps us to see that neither gender escapes stigma intensified by misogyny. Though men generally have more power to change or maintain the status quo, they also find themselves at the receiving end of stigma created and perpetuated by hegemonic patriarchy. These standards are particularly debilitating for MSM and transgender individuals in sub-Saharan Africa who do not conform to masculine cultural ideals.

## Implications

To make strides in HIV or TB prevention and care efforts, we must address barriers to infectious disease testing and treatment adherence, with renewed consideration for the unique concerns of men in sub-Saharan Africa. Researchers and medical providers have widely accepted undetectable equals untransmissible (U = U) messaging as an essential component of HIV elimination. However, U = U is predicated on earlier successes in the care continuum, including diagnosis, linkage, and retention in care, all of which are potentially compromised by stigma. Heterosexual HIV transmission remains the biggest driver of the epidemic in Sub-Saharan Africa [32]. If we become so focused on the intersection of women and HIV that we allow stigma to prevent male engagement in the care cascade, then we will never halt heterosexual transmission.

The knowledge gained in this review supports innovative and discrete means of testing to engage men in infectious disease care. Men avoid testing due to threats to their masculine identity. Low barrier testing and treatment models should accommodate work schedules and prioritize confidentiality and privacy [33] which were both highlighted by participants as essential for care engagement. Same day initiation strategies are essential to preserving physical strength and earning potential [34]. Men may be more receptive to home-based testing initiatives, mobile clinics, and education about the utility of ART and TB treatments for preserving strength and earning capacity [35]. From a structural standpoint, engaging men will require formal employment protections that prevent employment discrimination and allow reasonable accommodation for healthcare attendance [34].

Due to economic insecurity and potential for violence following HIV status disclosure, healthcare providers should incorporate interpersonal violence and housing instability screening into HIV testing and treatment visits for women. To support motherhood identity, health care providers must reinforce U = U education at all points of healthcare access and ensure that women know this also applies to transmission between mother and child. Similarly, women will benefit from education about the benefits of ART and TB treatment in the restoration of physical health [36]. Messaging centered around ART and healthy weight may be an effective strategy combating internalized and anticipated stigma. Furthermore, bolstering female economic opportunities leads to empowerment in the home and in healthcare decision making [37]. In the few articles that considered female economic empowerment, there was little discussion around how income changed female or motherhood identity and the long-term social consequences of that identity change. The impact of female financial enfranchisement is certainly an intervention area for scale up given its potential for stigma reduction.

The conceptual framework that has emerged from this meta-synthesis has the potential to guide stigma intervention and mitigation strategies that center socially relevant identities. Using the stigma identity framework, community level stigma arises from individuals' tendency to categorize and differentiate from others based on divergent characteristics. Stigma arises when individuals assess, judge, and differentiate from others. To address stigma at the individual level, healthcare providers must acknowledge the unique identity of their patients and how their patients fit into their nuanced social positions. Without this acknowledgment, healthcare providers will misunderstand the motivations behind seemingly negative healthcare behaviors. An underappreciation of cultural nuance and personal circumstances leads healthcare providers to dismiss patients as "non-compliant" or "defaulters". When these terms are used, patients report feeling dehumanized and are allowed to slip into a medically constructed narrative rife with its own healthcare-specific biases.

Throughout this analysis it was clear that healthcare providers continue to enact stigma in their daily interactions with patients. Despite an acknowledgment that stigma is a barrier to care and should be addressed, healthcare providers openly endorsed stigma towards substance use disorders, mental illness and LGBTQ identifying individuals. Healthcare providers in HIV and TB care have come far in their treatment of PLWH but must be cognizant of similar biases towards addiction, mental illness, and same sex behavior. Education and empathy remain central to stigma mitigation in healthcare. Without careful self-reflection, healthcare providers will be unable to engage patients throughout the HIV care continuum, especially longstanding key populations who oftentimes are more acutely subjected to intersectional stigma.

## Conclusion

Individual studies of stigma interrogate the role of intersectionality and an infinite combination of traits or characteristics that affect the internalization, experience, and perception of stigma. However, less studied is the impact of multiple identities on stigma and stigma on multiple identities. From our analysis, gender and HIV status does not create a simple intersection. Gender is a role and an identity central to social and self-actualization. This reality elevates gender, adulthood, parenthood, and occupational identity from social determinants of health to core constructs central to human psychosocial fulfillment in the sub-Saharan context. These socially constructed identities actively created and re-shaped by culture and individuals. Throughout the history of HIV and AIDS in the region, cultural identity remains a powerful force that dictates behavior and health outcomes. Future stigma research should consider not only the barriers and difficulties created by these intersections, but also how these intersections may be central to the identity of individuals.

## Supporting information

**S1 Table. Sample search strategy–PubMed.**
(DOCX)

**S1 Text. Article includes in this metasynthesis.**
(DOCX)

## Acknowledgments

We would like to acknowledge Ms. Stella Seal, Johns Hopkins University informationist for her assistance in the search strategy. We would also like to acknowledge the authors of the original manuscripts for their dedication to this topic and for elevating the voices of marginalized people in sub-Saharan Africa.

## Author Contributions

**Conceptualization:** Alanna J. Bergman, Dalmacio Dennis Flores.

**Formal analysis:** Alanna J. Bergman, Katherine C. McNabb, Khaya Mlandu, Alvine Akumbom.

**Methodology:** Alanna J. Bergman, Dalmacio Dennis Flores.

**Supervision:** Dalmacio Dennis Flores.

**Writing – original draft:** Alanna J. Bergman.

**Writing – review & editing:** Alanna J. Bergman, Katherine C. McNabb, Khaya Mlandu, Alvine Akumbom, Dalmacio Dennis Flores.

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
