## [Decision Letter · Decision Letter 0]

1 Jun 2022

PGPH-D-22-00449

Identity Management in the Face of Intersecting Stigmas: A Metasynthesis of Qualitative Reports from sub-Saharan Africa

Dear Author.

Thank you for submitting your manuscript to PLOS Global Public Health. After careful consideration, we feel that it has merit but does not fully meet PLOS Global Public Health’s publication criteria as it currently stands. Therefore, we invite you to submit a revised version of the manuscript that addresses the points raised during the review process.

We look forward to receiving your revised manuscript.

Kind regards,

Maria del Mar Pastor Bravo, Ph.D.

Academic Editor

Journal Requirements:

a. Please clarify all sources of funding (financial or material support) for your study. List the grants (with grant number) or organizations (with url) that supported your study, including funding received from your institution. 

b. State the initials, alongside each funding source, of each author to receive each grant.

c. State what role the funders took in the study. If the funders had no role in your study, please state: “The funders had no role in study design, data collection and analysis, decision to publish, or preparation of the manuscript.”

d. If any authors received a salary from any of your funders, please state which authors and which funders.

2. Please ensure that the funders and grant numbers match between the Financial Disclosure field and the Funding Information tab in your submission form. Note that the funders must be provided in the same order in both places as well.

3. Please update your Competing Interests statement. If you have no competing interests to declare, please state: “The authors have declared that no competing interests exist.”

4. Please amend your Data Availability Statement and indicate where the data may be found.

Additional Editor Comments (if provided):

Reviewers' comments:

Reviewer's Responses to Questions

**Comments to the Author**

1. Does this manuscript meet PLOS Global Public Health’s publication criteria? Is the manuscript technically sound, and do the data support the conclusions? The manuscript must describe methodologically and ethically rigorous research with conclusions that are appropriately drawn based on the data presented.

Reviewer #1: Yes

Reviewer #2: Yes

2. Has the statistical analysis been performed appropriately and rigorously?

Reviewer #1: N/A

Reviewer #2: N/A

3. Have the authors made all data underlying the findings in their manuscript fully available (please refer to the Data Availability Statement at the start of the manuscript PDF file)?

Reviewer #1: Yes

Reviewer #2: Yes

4. Is the manuscript presented in an intelligible fashion and written in standard English?

Reviewer #1: Yes

Reviewer #2: Yes

5. Review Comments to the Author

Reviewer #1: The article is well written and elucidates a rigorous effort made by the authors to do justice with this study.

I have only two major comments on the article:

1. Even though the majority of the article focuses on HIV and associated stigma, it is not reflected in the title.

2. This paper will benefit if the definitions and concepts behind “Stigma” and “Intersectionality” are laid out first.

Thanks for the amazing job.

Best,

Reviewer #2: Thank you for the opportunity to review this manuscript. It is an interesting application of a qualitative metasynthesis approach to examine intersecting stigmas. My comments are mainly requests for clarifications, details and further reflection.

Title & abstract

1) The focus is on HIV but this is not mentioned in the title. It may be relevant to specify given that it is included in your aim.

Introduction

2) I can see that you have already included a brief justification of the geographical focus of the metasynthesis but why is it framed in contrast with North American contexts? I would love to see a bit more transparent reflection on this regional focus, even if some of the reasons may be purely pragmatic (e.g. feeds into or informs a specific project). Not all gaps in knowledge or evidence need to be filled either, so if there is a way to briefly explain why synthesising evidence from this region is particularly relevant (other than it not being done before) then that would be even more helpful to the readers.

Methods

3) Were you open to any infectious disease stigma, or was HIV an inclusion criterion? It is not mentioned as one but it is interesting that all studies addressed it. This also relates to the next comment about search strategy.

4) It would be useful to either see one or two full examples of the search strategies (perhaps a table or text box), or get a bit more details about how they differed, e.g. by database or what does it mean that a “variety” was used?

5) The numbers in the PRISMA diagram do not seem to fully add up so some clarification or correction is needed.

Results

6) Is “same-sex behaviour” the best term to use here? If you are referring to sexual behaviour, then the word ‘sexual’ is probably missing. It also seems to me that this term is often used in animal studies so I would just make sure that it is considered appropriate for this context.

7) Your synthesis is very interesting but it is difficult to trace the evidence from the way it is presented with few references. Is there a reason why the supplementary tables cannot be part of the paper itself? I would be in favour of that if journal guidelines allow it as that would help the reader see which studies informed which part of the synthesis.

8) Line 354 typo: “threated”

Discussion & conclusion

9) Line 366 “as” missing from “such as”

10) I am a bit uncomfortable with how the whole of Sub-Saharan Africa is lumped together in explanations about traditional masculinities, can you add some nuance here? Again, being able to trace the evidence would also help as the reader would see where specific statements come from.

6. PLOS authors have the option to publish the peer review history of their article (what does this mean?). If published, this will include your full peer review and any attached files.

**Do you want your identity to be public for this peer review?** For information about this choice, including consent withdrawal, please see our Privacy Policy.

Reviewer #1: No

Reviewer #2: No

---

## [Decision Letter · Decision Letter 1]

12 Dec 2022

Identity Management in the Face of HIV and Intersecting Stigmas: A Metasynthesis of Qualitative Reports from sub-Saharan Africa

PGPH-D-22-00449R1

Dear Ms McNabb,

We are pleased to inform you that your manuscript 'Identity Management in the Face of HIV and Intersecting Stigmas: A Metasynthesis of Qualitative Reports from sub-Saharan Africa' has been provisionally accepted for publication in PLOS Global Public Health.

Best regards,

Julia Robinson

Executive Editor

Reviewer Comments (if any, and for reference):

Reviewer's Responses to Questions

**Comments to the Author**

1. If the authors have adequately addressed your comments raised in a previous round of review and you feel that this manuscript is now acceptable for publication, you may indicate that here to bypass the “Comments to the Author” section, enter your conflict of interest statement in the “Confidential to Editor” section, and submit your "Accept" recommendation.

Reviewer #1: All comments have been addressed

Reviewer #2: All comments have been addressed

2. Does this manuscript meet PLOS Global Public Health’s publication criteria? Is the manuscript technically sound, and do the data support the conclusions? The manuscript must describe methodologically and ethically rigorous research with conclusions that are appropriately drawn based on the data presented.

Reviewer #1: Yes

Reviewer #2: (No Response)

3. Has the statistical analysis been performed appropriately and rigorously?

Reviewer #1: N/A

Reviewer #2: (No Response)

4. Have the authors made all data underlying the findings in their manuscript fully available (please refer to the Data Availability Statement at the start of the manuscript PDF file)?

Reviewer #1: Yes

Reviewer #2: (No Response)

5. Is the manuscript presented in an intelligible fashion and written in standard English?

Reviewer #1: Yes

Reviewer #2: (No Response)

6. Review Comments to the Author

Reviewer #1: Dear Authors,

Thank you for revising the manuscript per the peer review comments.

The manuscript looks sound and will be significant contribution to the global pool of literature.

I am happy to recommend the manuscript for publication.

Thanks

Reviewer #2: Thank you for your engagement with my comments - I am happy with how they have been addressed.

7. PLOS authors have the option to publish the peer review history of their article (what does this mean?). If published, this will include your full peer review and any attached files.

**Do you want your identity to be public for this peer review?** For information about this choice, including consent withdrawal, please see our Privacy Policy.

Reviewer #1: **Yes: **Ateeb Ahmad Parray

Reviewer #2: No
